# Impacts of Climate Change on the Precipitation and Streamflow Regimes in Equatorial Regions: Guayas River Basin



Mercy Ilbay-Yupa [1,*], Franklin Ilbay [2], Ricardo Zubieta [3], Mario García-Mora [4] and Paolo Chasi [1]

1   Facultad de Ciencias Agropecuarias y Recursos Naturales, Universidad Técnica de Cotopaxi (UTC), Latacunga 050150, Ecuador; wilman.chasi@utc.edu.ec
2   Programa de Maestría en Ciencias con Mención en Hidráulica, Universidad Nacional de Ingeniería, Lima 15012, Peru; luis_5840@hotmail.com
3   Subdirección de Ciencias de la Atmósfera e Hidrósfera (SCAH), Instituto Geofísico del Perú (IGP), Lima 15012, Peru; rzubieta@igp.gob.pe
4   Carrera de Tecnología Superior en Floricultura, Instituto Superior Tecnológico Cotopaxi, Latacunga 050150, Ecuador; gvictormario@hotmail.es
*   Correspondence: mercy.ilbay@utc.edu.ec

**Abstract:** The effects of climate change projected for 2050 to 2079 relative to the 1968–2014 reference period were evaluated using 39 CMIP5 models under the RCP8.5 emissions scenario in the Guayas River basin. The monthly normalized precipitation index (SPI) was used in this study to assess the impact of climate change for wet events and droughts from a meteorological perspective. The GR2M model was used to project changes in the streamflow of the Daule River. The climate projection was based on the four rigorously selected models to represent the climate of the study area. On average, an increase in temperature (~2 °C) and precipitation (~6%) is expected. A 7% increase in precipitation would result in a 10% increase in streamflow for flood periods, while an 8% decrease in precipitation could result in approximately a 60% reduction in flow for dry periods. The analysis of droughts shows that they will be more frequent and prolonged in the highlands (Andes) and the middle part of the basin. In the future, wet periods will be less frequent but of greater duration and intensity on the Ecuadorian coast. These results point to future problems such as water deficit in the dry season but also increased streamflow for floods during the wet season. This information should be taken into account in designing strategies for adaptation to climate change.

**Keywords:** climate change; droughts; Guayas; Ecuador; impacts

## 1. Introduction

Climate change is one of the main threats to the planet and constitutes a challenge for the sustainable management of water resources [1,2]. It also plays an important role in social development and global economic growth. In the future, changes in the spatial and temporal patterns of rainfall are expected, leading to an increase in extreme weather events [3–6]. These impacts are also expected to manifest in hydrological regimes, which influence the spatial and temporal patterns of water resources [7,8]. All these climate projections were derived from a variety of climate change models [3,9–12]. Global climate models (GCMs) are mathematical models that integrate components of the Earth, atmospheric and ocean climate systems according to physical, chemical and biological principles to simulate climate with respect to $CO_2$ emission conditions [13]. GCMs are generally used to understand the current climate and project climate change, allowing robust decisions to be made based on a wide range of possible futures [14]. The reliability of scenarios generally depends on their ability to represent climate processes [15], the type of emission scenario [16] and model structure [17]. The reliability of some GCMs and regional circulation models in equatorial regions, as a first approximation, has already been characterized by [18].

The impacts of droughts in an area are assessed based on changes of frequency, duration and intensity [19]. There are several studies that have analyzed meteorological droughts in relation to cumulative precipitation shortages during wet season onset [20]. Droughts characterized can severely alter the hydrologic regime [21], affect river water quality and impact aquatic ecosystems [22]. The severity and frequency of climatic events such as droughts and floods will change in the future due to global warming, severely affecting the natural environment [23]. Increases in rainfall can cause events such as floods and landslides, among others [24]. Floods are very costly natural disasters [25] that strongly affect the population and economic development, leading to increased vulnerability [26].

Land use changes and climate change are dominant processes that affect hydrological processes such as evapotranspiration, interception and infiltration, resulting in alterations of surface and groundwater flows [27,28]. The availability of water resources is decreasing due to population growth [29]. Understanding changes in the course of climate would make it possible to design more effective policies for adaptation in a changing climate [30]. Developed countries, which are the main contributors of greenhouse gases, should focus on mitigation, while developing countries should work on adaptation.

Ecuador is a country that has experienced climatic changes, such as increased temperatures and rainfall, which have had visible impacts, including the melting of glaciers. Increases in average annual temperature and precipitation of 2–3 °C and 3%, respectively, are expected for the period 2030–2049 [31]. This represents substantially higher temperature increases than the global average [32]. However, there are few basin-scale studies in equatorial regions that provide crucial local details and information for management, prudent environmental protection and planning of economic and social activities [33]. The Guayas River basin is one of the most important in Ecuador and is home to approximately 40% of the population. This area has been impacted by severe and prolonged extreme events that have had destructive effects on the economy, food security, infrastructure and ecosystems [34–36]. The objective of this research was to characterize the impacts of climate change on the dynamics of precipitation, temperature and streamflow in the Guayas River basin.

## 2. Materials and Methods

### 2.1. Study Area

The Guayas River basin is located in the central–western part of Ecuador and is one of the country's main river basins. It covers an area of approximately 32,890 km$^2$ and drains into the Gulf of Guayaquil, with an average annual discharge of 200 m$^3$/s for the dry season, increasing to 1600 m$^3$/s during the wet season [37]. The two main rivers are the Daule and Babahoyo, which merge into the Guayas River near Guayaquil, the largest and most populated city in Ecuador, with an area of 1800 km$^2$ and a population of 3.645 million [38].

Average annual precipitation is around 1800 mm, and 89% of total precipitation occurs during the wet season, from December to May [36,39]. The basin has two homogeneous precipitation regions. The first is located in the high mountain zone and is classified as a region of low erosivity. The second, the coastal region, is a region of high aggressiveness, with greater erosive potential and sedimentation problems in the lower part of the basin due to the erosive capacity of rainfall [40]. The Guayas River basin has been affected by extreme phenomena such as El Niño [41], which caused flooding in 1965, 1972, 1973, 1976, 1983, 1987, 1992 and 1998 [36,39,42], and La Niña [43]. The lower part of the basin has greater exposure to flooding [44]. This basin contains the Daule–Peripa reservoir, with an area of ≈30,000 ha, 6 billion m$^3$ of water storage capacity and discharge of 14,350 m$^3$/s, which supplies drinking water to Guayaquil and cities bordering the Daule River. The reservoir was built in 1987 to generate electricity, supply water for irrigation, control floods and provide drinking water [45]. The basin has high agricultural potential; its main products include subsistence and export crops, livestock and shrimp [36].

## 2.2. Data

### 2.2.1. Meteorological and Hydrological Data

The database includes monthly rainfall and temperature records from meteorological stations and a hydrological station managed by INAMHI (National Meteorological and Hydrological Institute of Ecuador) (Table 1). The selected stations contain information from at least 30 years during the period 1968–2014. There are 16 stations located at altitudes below 1500 masl (Pacific Coast), and 11 stations are in the mountain region (Figure 1). A rigorous quality check of these data was performed using the regional vector method. More details about this analysis can be found in [46].

### 2.2.2. GCMs Data

Monthly temperature and precipitation variables simulated from 39 models of the Coupled Model Intercomparison Project 5 (CMIP5), under the high emission concentration scenario RCP8.5, were used. These are available online (https://climexp.knmi.nl/start.cgi, accessed on 2 October 2020) for the period 1850 to 2100. These models provide a series of simulated future climate variables that characterize the coming decades or centuries. These can be used as a basis for exploring the impacts of climate change on policy issues of interest and relevance to society [10]. To highlight the largest impacts of climate change (i.e., considering the largest change in temperature, radiative forcing (>8.5 W/m$^2$) and concentrations (>1370 $CO_2$ ppm) for 2100 [47], the Representative Concentration Pathway 8.5 (RCP8.5) scenario was selected. This RCP has also been used to evaluate climate change in droughts [48] and hydroelectric power generation [49] in Ecuador.

**Table 1.** Meteorological stations and hydrology of the Guayas River basin.

| Name | Latitude (°S) | Longitude (°W) | Altitude (masl) | Period | | |
|---|---|---|---|---|---|---|
| | | | | Precipitation | Temperature | Streamflows |
| Salinas–Bolívar | −1.40 | −79.02 | 3600 | 1968–2014 | | |
| Achupallas–Chimborazo | −2.28 | −78.77 | 3178 | 1968–2014 | | |
| Pangor–J.de Velasco | −1.83 | −78.88 | 3109 | 1969–2014 | | |
| Cañi–limbe | −1.77 | −78.99 | 2800 | 1977–2014 | | |
| Guasuntos | −2.23 | −78.81 | 2438 | 1972–2014 | | |
| Compud | −2.34 | −78.94 | 2402 | 1968–2014 | | |
| Chillanes | −1.98 | −79.06 | 2330 | 1968–2014 | 1982–2014 | |
| Alausi | −2.2 | −78.85 | 2267 | 1968–2014 | | |
| San Antonio–Monjas River | −1.58 | −79.13 | 2200 | 1980–2014 | | |
| Chunchi | −2.28 | −78.92 | 2177 | 1968–2014 | 1982–2014 | |
| Pallatanga | −2.00 | −78.97 | 1523 | 1968–2014 | | |
| Ramón Campaña | −1.12 | −79.09 | 1462 | 1968–2014 | | |
| Chimbo Pj Pangor | −1.94 | −79.00 | 1452 | 1968–2014 | | |
| Sto. Domingo Airport | −0.25 | −79.20 | 554 | 1968–2098 | | |
| Bucay | −2.20 | −79.13 | 480 | 1968–2000 | | |
| Las Delicias–Pichincha | −0.26 | −79.40 | 340 | 1968–2003 | | |
| Puerto Ila | −0.48 | −79.34 | 319 | 1968–2014 | 1970–2014 | |
| Echeandia | −1.43 | −79.29 | 308 | 1968–2014 | | |
| San Juan La Mana | −0.92 | −79.25 | 215 | 1968–2014 | | |
| Colimes de Pajan | −1.58 | −80.51 | 200 | 1970–2014 | | |
| Camposano #2 | −1.59 | −80.40 | 113 | 1977–2014 | 1982–2014 | |
| Pichilingue | −1.07 | −79.49 | 81 | 1968–2014 | 1978–2014 | |
| Ingenio San Carlos | −2.22 | −79.41 | 63 | 1968–2014 | | |
| Milagro (Ingenio Valdez) | −2.12 | −79.60 | 23 | 1968–2014 | 1970–2014 | |
| Pueblo Viejo | −1.52 | −79.54 | 19 | 1968–2014 | 1984–2014 | |
| Vinces INAMHI | −1.56 | −79.77 | 14 | 1968–2014 | | |
| La Capilla INAMHI | −1.70 | −80.00 | 7 | 1968–2014 | | |
| Daule en la Capilla | 1.69 | 79.99 | 13 | | | 1982–2014 |

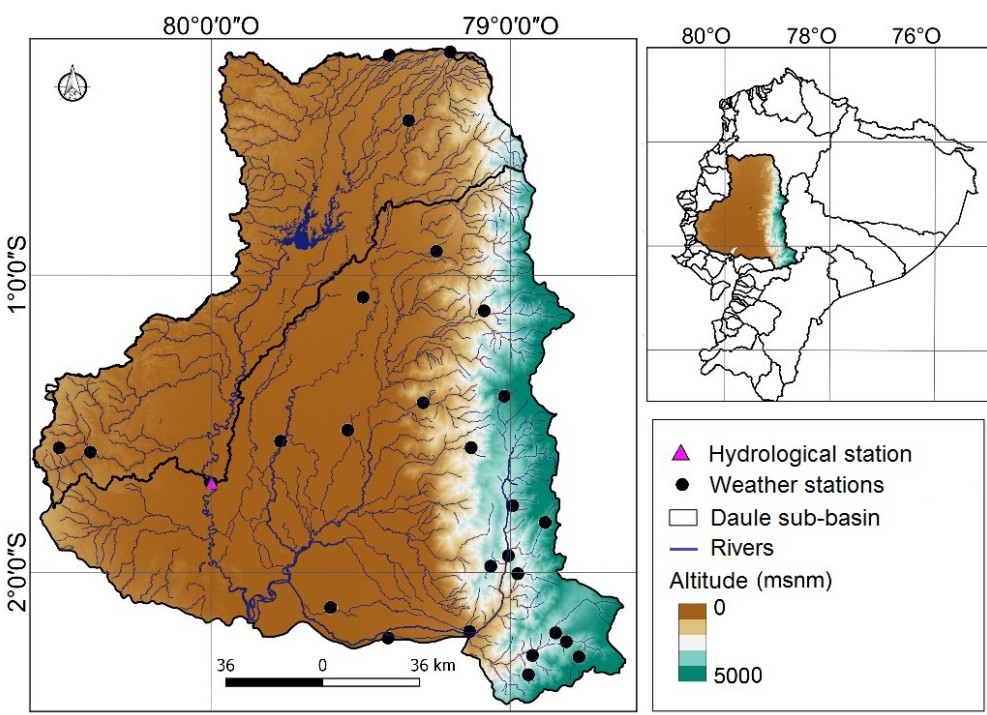

**Figure 1.** Location of the Guayas River basin, showing altitude and distribution of meteorological and hydrological stations.

*2.3. Methodology*

The methodology comprised four phases: (i) Collection, processing and analysis of temperature, observed precipitation and discharge data for the 1968–2014 (baseline scenario) and 2050–2079 (future) GCMs in the Guayas River basin. (ii) Correction and validation of monthly data for the 39 GCMs. (iii) Analysis of delta changes, wet and drought events (SPI) and hydrological modeling (GR2M). (iv) Determining the impacts of climate change (Figure 2).

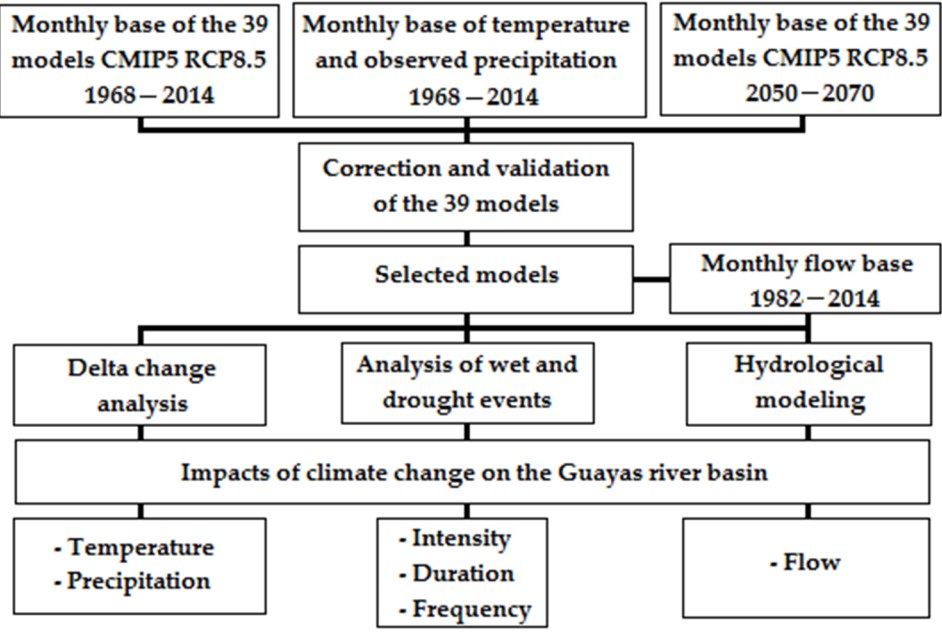

**Figure 2.** Diagram of the methodological phases of the climate change impact assessment in the Guayas River basin.

2.3.1. Assessment of GCMs

Future predictions in the basin were based only on the models that were able to simulate the observed characteristics of the climate system. For this purpose, the 39 GCMs were evaluated using the Taylor diagram. This design provides a concise statistical summary of the patterns that coincide with the correlation coefficient (R), root mean square difference (RMSE) and the ratio of their variances. These statistics facilitate the determination of how much of the overall RMSE difference in the patterns is attributable to a difference in variance and how much is due to poor pattern correlation of observed to simulated values, based on Equations (1) and (2) [48]. The plot is widely used to evaluate or to track changes in the performance of complex models such as geophysical phenomena [23,49–51].

$$\text{RMSE}^2 = \sigma_f^2 + \sigma_r^2 - 2\sigma_f \sigma_r R \tag{1}$$

where $\sigma_f^2$ and $\sigma_r^2$ are the variances of the test and reference fields, respectively. The construction of the diagram is based on the similarity of the above Equation and the Cosine Law:

$$c^2 = a^2 + b^2 - 2ab\ \cos\varnothing \tag{2}$$

The selection of the best temperature and precipitation GCMs in comparison to observed dataset was performed using the skill score (*S*) for each model. The skill score varies from zero (least skillful) to one (most skillful) and is defined as shown in Equation (3) [48]:

$$S = \frac{4\,(1+R)^4}{\left(\hat{\sigma}_f + 1/\hat{\sigma}_f\right)^2 (1+R_0)^4} \tag{3}$$

where $S$ = ability score of the models; $R$ = Pearson's correlation coefficient; $\hat{\sigma}_f = \sigma_f/\sigma_r$ = normalized standard deviation; and $R_0$ = maximum correlation achievable.

2.3.2. Precipitation and Temperature Correction

The adjustment between the observed and simulated (base) monthly mean data during the historical period (1968–2014) was performed using the monthly mean data correction [52], where the temperature is obtained by adding to the entire time series a constant displacement called "C", which is equal to the average difference between observations and simulations during the reference period (Equation (4)). It is important to note that for average monthly temperature of the basin, only information from 1982 to 2014 was used, as that was the period for which data were available.

$$\widetilde{T}_{GCM\ Bas.} = C + T_{GCM\ Bas.} \tag{4}$$

where $\widetilde{T}_{GCM\ Bas.}$ and $T_{GCM\ Bas.}$ are the corrected and uncorrected base temperatures of the models, respectively. The coefficient "C" is obtained as follows:

$$C = \frac{\sum_{i=1}^{n} T_{Obs.} - \sum_{i=1}^{n} T_{GCM\ Bas.}}{n} \tag{5}$$

where $T_{Obs.}$ is the observed temperature in the Guayas River basin, and $n$ is the number of years of analysis.

The mean monthly precipitation was obtained by means of a multiplicative factor differentiated for each month (Equation (6)):

$$\widetilde{P}_{GCM\ Bas.} = C * P_{GCM\ Bas.} \tag{6}$$

where $\widetilde{P}_{MCG\ Bas.}$ and $P_{MCG\ Bas.}$ are the base, corrected and uncorrected rainfall of the models, respectively. The coefficient "C" is obtained with the following Equation:

$$C = \sum_{i=1}^{n} P_{Obs.} / \sum_{i=1}^{n} P_{GCM\ Bas.} \tag{7}$$

where $P_{Obs.}$ is the precipitation observed in the Guayas River basin.

Subsequently, the performance of the 39 monthly models was evaluated with the observed data for the temperature and precipitation variables, using the Taylor diagram as indicated above.

### 2.3.3. Delta Change Analysis

This analysis is conducted by transferring the mean monthly change between the GCMs for the base period (1968–2014) and future scenario (2050–2079) to the series observed in the basin stations (1968–2014) for the temperature, precipitation and evapotranspiration variables, using the Equations proposed by [53]:

$$T_{Fut.} = T_{Obs.} + \left( \overline{T_{GCM\ Bas.}} - \overline{T_{GCM\ Fut.}} \right) \tag{8}$$

$$P_{Fut.} = P_{Obs.} * \left( \overline{P_{GCM\ Fut.}} / \overline{P_{GCM\ Bas.}} \right) \tag{9}$$

where $T_{Obs.}$ and $P_{Obs.}$ are monthly temperatures and precipitation observed in the basin; $\overline{T_{GCM\ Bas.}}$ and $\overline{P_{GCM\ Bas.}}$ are base monthly temperatures and precipitation of the GCMs; and $\overline{T_{GCM\ Fut.}}$ and $\overline{P_{GCM\ Fut.}}$ are future monthly temperatures, precipitation and evapotranspiration of the GCMs.

### 2.3.4. Wet Periods and Drought Analysis

The standardized precipitation index (SPI) at a monthly time scale (SPI-1), which is based on precipitation datasets [54], was used for 27 observed precipitation time series for the observed period (1968–2014) and future scenario period (2050–2079) of the 39 GCMs. The SPI is based on the conversion of precipitation data to probabilities using long-term monthly precipitation records calculated on different time scales. SPI is a flexible index for estimating short- and long-term droughts for agricultural and hydrological applications [55–58]. SPI is calculated by fitting an appropriate probability density function to the summed precipitation frequency distribution on the time scale of interest (e.g., 1, 3, 6 and 12 months).

Once the monthly SPI values were determined, the severity of droughts and wet scenarios were evaluated using Table 2.

**Table 2.** Classification of droughts by SPI value, according to [59].

| SPI Value | Category |
|---|---|
| 2.00 or more | Extremely wet |
| 1.50 a 1.99 | Severely wet |
| 1.00 a 1.49 | Moderately wet |
| 0 a 0.99 | Mildly wet |
| 0 a −0.99 | Mild drought |
| −1.00 a −1.49 | Moderate drought |
| −1.50 a −1.99 | Severe drought |
| −2 or less | Extreme drought |

The changes in drought regime were characterized by the frequency, duration and intensity of droughts (moderate, severe and extreme) computed from the SPI-1 [3,60,61]. In this article, a dry month is considered when the SPI-1 was less than or equal to the −1.0 threshold. The frequency is referred as the total number of dry months over a defined period. Drought intensity was defined as the average value of the SPIs below the threshold over a defined period, while the duration of a drought event was defined as the consecutive and uninterrupted time period having an SPI-1 below the threshold over a defined period. The changes of drought regime (duration, intensity and frequency) were analyzed for the 2050–2079 period in relation to 1968–2014 period.

### 2.3.5. Hydrological Modeling by GR2M

Hydrological modeling with GR2M [62] could not be performed for the entire basin due to the lack of information on the flow of the Guayas River. It was performed for the middle and upper sub-basin of the Daule River, however, using data from the Daule at the Capilla hydrological station (Figure 1). The modeling period was 33 years (1982–2014), because of the extent of the temperature data. GR2M is a rainfall–runoff model at a monthly time step, which uses input data such as precipitation and evapotranspiration. Its structure is based on two calibration parameters and two reservoirs: (i) the quadratic soil reservoir $S$ (soil reservoir), which defines the production function with a maximum capacity $X1$; and (ii) the gravity reservoir $R$ (water reservoir), which specifies the transfer function with $X2$ (second parameter). $X2$ determines runoff on output ($Q$) and the water exchange between surface and subsurface processes [63]. This model has been used in an Andean basin south of Ecuador and adequately represents flow dynamics but overestimates low flows [64].

The GR2M calculation is produced as an output of Equation (10), where the model simulates a complete hydrologic equilibrium [65]:

$$P = Q + E + F + \Delta S \tag{10}$$

where $P$ is precipitation, $Q$ is total runoff, $E$ is evapotranspiration, $F$ is total groundwater recharge and $\Delta S$ is the change in soil water content. All components are expressed in mm/yr. In this study, the GR2M model was divided into two periods: calibration, corresponding to the period 1982–2001, and validation (2002–2014), based on double or split sampling.

Previously, to estimate evapotranspiration time series, a method based on the average monthly temperature of the basin provided by [66] was used. This model is an efficient simplified method based on temperature, latitude and radiation [66,67]; the model formula is:

$$E = \frac{R_a \, (T_a + 5)}{100 \, \lambda \, \rho} \tag{11}$$

where $E$ is in mm/d, $R_a$ is the shortwave extraterrestrial radiation (kJ/m$^2$/d), $T_a$ is the air temperature (°C), $\lambda$ is the latent heat of vaporization (kJ/kg) and is the density of water (kg/L). This method is used to determine evapotranspiration in wet areas, because it provides better estimates in those climatic conditions than in arid areas [68–70].

Finally, the criteria for validation of the GR2M model for the two samples (calibration and validation) were the Nash coefficient and the coefficient of determination. The Nash criterion [71] compares the root mean square deviation of the flow roots with the variance, according to the following formula:

$$Nash = 100 \left[ 1 - \left( \frac{\sum_{l=1}^{N} \left( \sqrt{Q_0} - \sqrt{Q_C} \right)^2}{\sum_{l=1}^{N} \left( \sqrt{Q_0} - \sqrt{Q_M} \right)^2} \right) \right] \tag{12}$$

where $Q_0$ is the observed flow; $Qc$ the flow simulated using the model; $Q_M$, average observed flows; and $N$ the observation number.

The coefficient of determination was calculated using the following Equation:

$$R^2 = \left[ \frac{Cov(x,y)}{\sigma_x . \sigma_y} \right]^2 \tag{13}$$

where $Q_0$ is the observed flow $\sigma_x$ and $\sigma_y$; $R^2$ is the coefficient of determination of calculated and observed flows; $\sigma_x$ and $\sigma_x$, the standard deviation of the flow series; $y$ is the calculated flow; and $x$ is observed flows. Monthly precipitation and temperature from the 39 models were used to project future streamflow.

# 3. Results

## 3.1. Model Performance

The 39 model datasets were evaluated with respect to the observed data from the control period (1968–2014). Performance before correction showed low correlation between the data observed in the basin and the GCM ($r$~0.4, $p \leq 0.05$) for the period 1968–2014, with standard deviation and RMSD higher than 190 mm/month and 110 mm/month respectively (Figure 3a). Similarly, the temperature data from GCMs without correction presented a correlation coefficient of less than 0.5 ($p \leq 0.05$), standard deviation higher than 0.44 °C/month and RMSD between 0.5–1.2 °C/month (Figure 3b). However, improved model performance was achieved after correction of the monthly mean data (red circles).

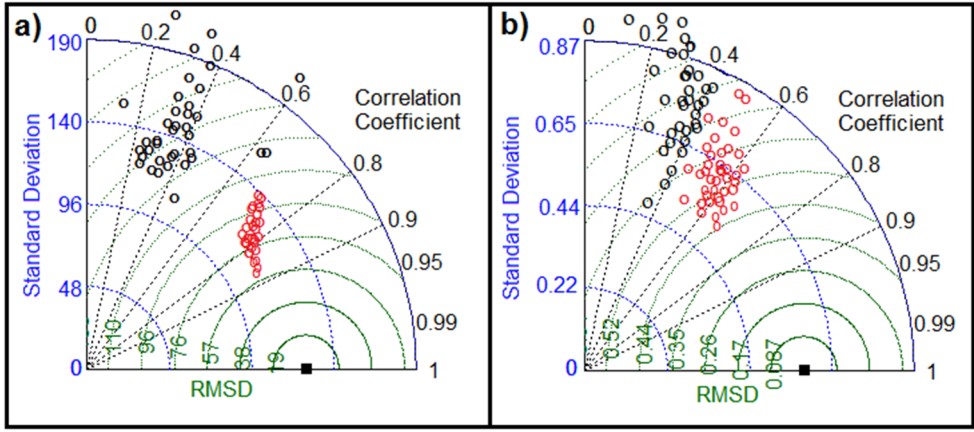

**Figure 3.** Performance of climate models with uncorrected (black circles) and corrected (red circles) precipitation (**a**) and temperature (**b**) monthly mean data using Taylor plot for the period 1968–2014.

The BNU-ESM, inmcm4, FGOALS-g2 and CSIRO-Mk3-6-0 models were selected for the precipitation and temperature datasets. These were chosen because they had the best skill scores (S $\geq$ 0.69). The GCMs selected in this work were also chosen to represent the observed characteristics of the climate system in America [72].

## 3.2. Analysis of Climatic Variables

Analysis of monthly mean precipitation and temperature using the corrected monthly mean data shows that the selected models represent very well the annual cycle of precipitation and temperature observed in the period 1968 to 2014 for the Guayas River basin (Figure 4); the distribution of precipitation is characterized by a wet season (December–May) and dry season (June–November).

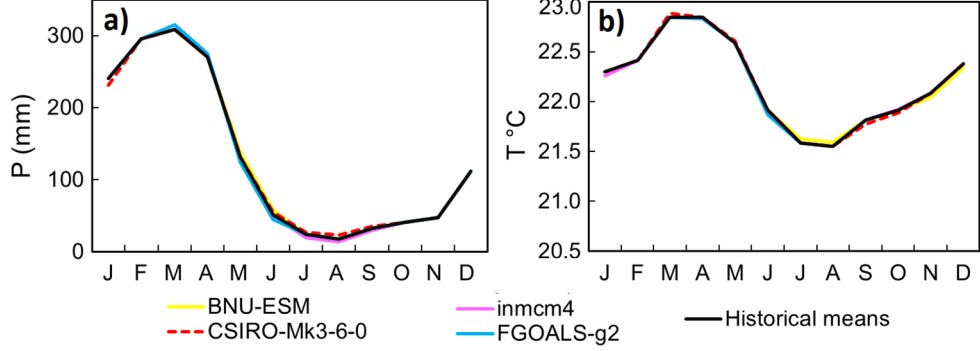

**Figure 4.** Comparison of monthly mean precipitation (**a**) and temperature (**b**) with the four selected climate models and observed data (1968–2014) for the Guayas River basin.

The average results of the four models predict a warmer climate for the basin, with increases of 1.9 to 2.5 °C for all months; May and August are the most affected (Figure 5a). Precipitation changes, however, are likely to be much more variable (Figure 5b). Under the RCP8.5 scenario, the four GCMs project average rainfall increases of between 0.3 to 14% monthly between November and July and decreases of between 0.3 and 10% between August and October (Figure 5b).

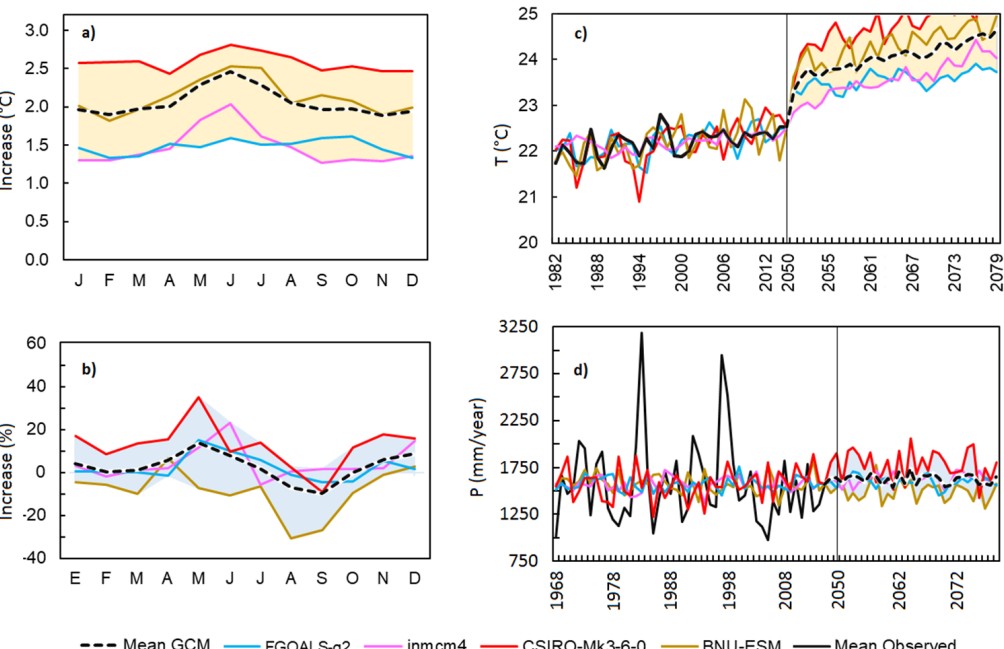

**Figure 5.** Monthly increases in temperature (**a**) and precipitation (**b**) and annual increases in temperature (**c**) and precipitation (**d**) projected for the Guayas River basin under four models of the RCP8.5 climate statistics for 2050–2079.

Annual temperature increases in the future (2050–2079) for all GCMs range from 1.3 to 2.6 °C (Figure 5c). Similarly, on average, the immcm4, FGOALS-g2 and CSIRO-Mk3-6-0 models predict an increase in annual precipitation (2–13%). However, only the BNU-ESM model predicts a decrease (5%) in the basin (Figure 5d). It is important to note that the interannual variability of temperature in the four models can represent the dynamics of the basin (Figure 5c). Nonetheless, the precipitation regimen is not adequately represented (Figure 5d).

### 3.3. Changes in Precipitation Characteristics

Figure 6 shows the cumulative distribution frequency curves (CDF) for drought events (SPI-1 $\leq -1$) lasting between 1 and 6 months and wet events (SPI-1 $\geq 1$) lasting between 1 and 4 months in the Guayas River basin. Changes in the duration of moderate, severe and extreme drought are found, with three of the four models projecting a decrease (Figure 6a). On average, a decrease of 5 (observed) to 4 months in droughts is expected. On average, wet periods will show a slight increase from 3 (observed) to 3.25 months, although three of the four models suggest they would maintain their duration (Figure 6b).

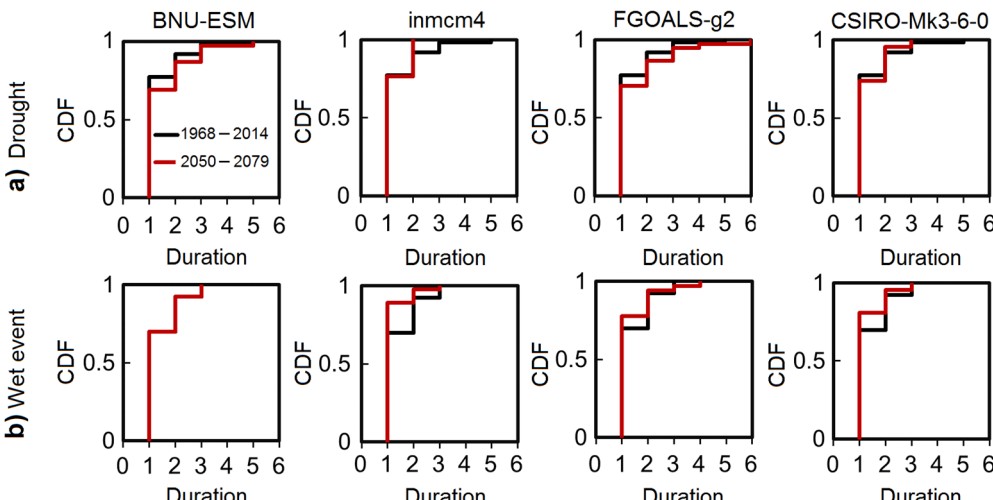

**Figure 6.** Cumulative distribution frequency (CDF) for drought periods lasting 1 to 6 months (**a**) and wet periods of 1 to 4 months (**b**) for the four climate models in the Guayas River basin during the observed (black line) and future (red line) periods.

Evaluation of the mean intensity of droughts (SPI-1 $\leq -1$) for the four selected GCMs shows an increase in intensity for all CDF curves (Figure 7a), while a decrease for the wet periods is also observed (SPI-1 $\geq 1$) (Figure 7b). Drought is expected to be significantly more intense, as SPI changes from $-2.45$ to $-2.97$ for the four selected GCMs. This is 1.21 times higher than the baseline. In contrast, on average, wet periods will be less intense (1.51 times lower than the observed line) for the four models selected.

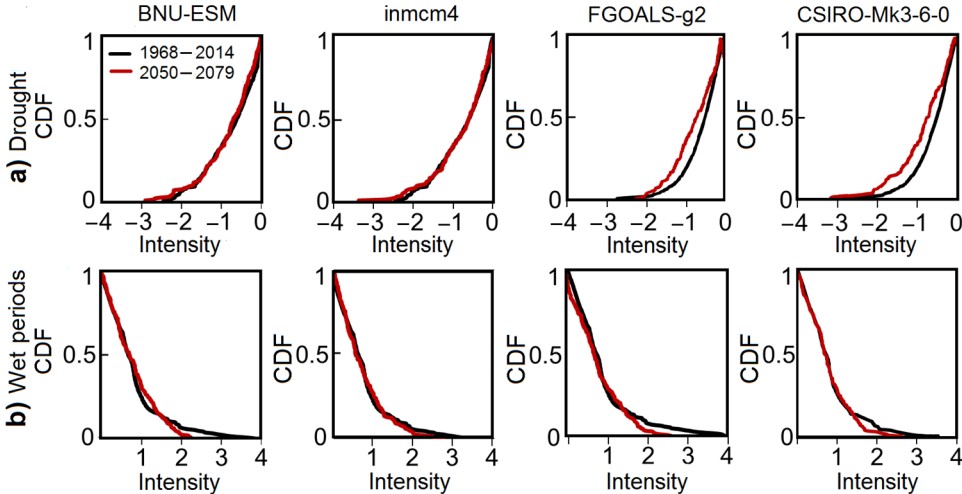

**Figure 7.** Cumulative distribution frequency curves (CDF) for the intensity of drought (between $-4$ and 0) (**a**) and wet (between 1 and 4) periods (**b**), from the four climate models in the Guayas River basin during the observed (black) and future (red) periods.

According to classification of droughts by SPI values (Table 2), on average, the four models show that, in the future, the intensity of extreme and severe droughts will increase slightly, while the intensity of mild and moderate droughts will decrease slightly. In contrast, for wet periods, the extremely wet category will decrease, while severe, moderate and mild wet periods will increase, on average, for 2050–2079 (Figure 8).

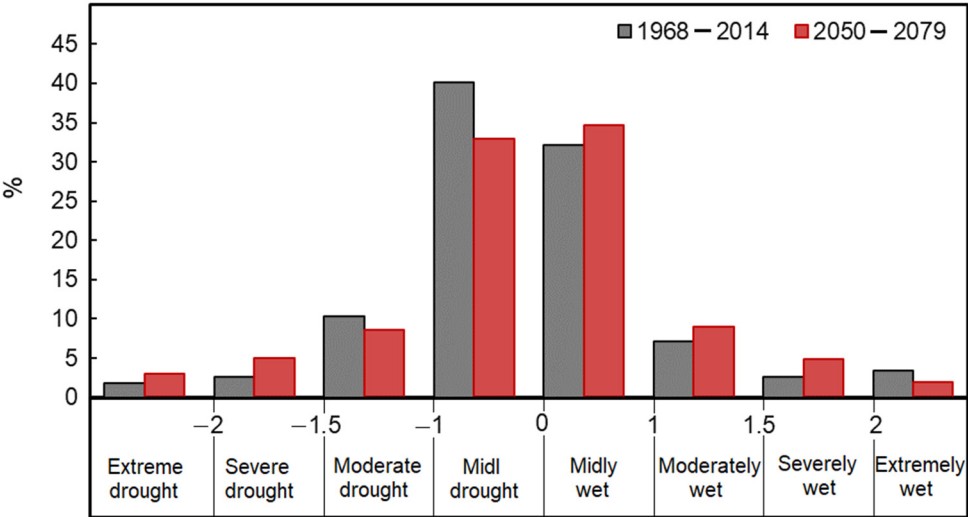

**Figure 8.** Classification of observed (bar) and future (black line) wet periods and droughts by SPI-1 value.

The spatial distribution of changes in the intensity, duration and frequency of droughts and wet periods SPI-1 in future climate scenarios (2050–2079) for the four selected models was estimated (BNU-ESM, inmcm4, FGOALS-g2 and CSIRO-Mk3-6-0) for the Guayas River basin (Figure 9). Figure 9a shows changes (%) in the duration of droughts. The plains of the Ecuadorian coast will show a decrease on average (~20%), while an increase (~5%) will be seen on the slopes of the western Andes Mountains.

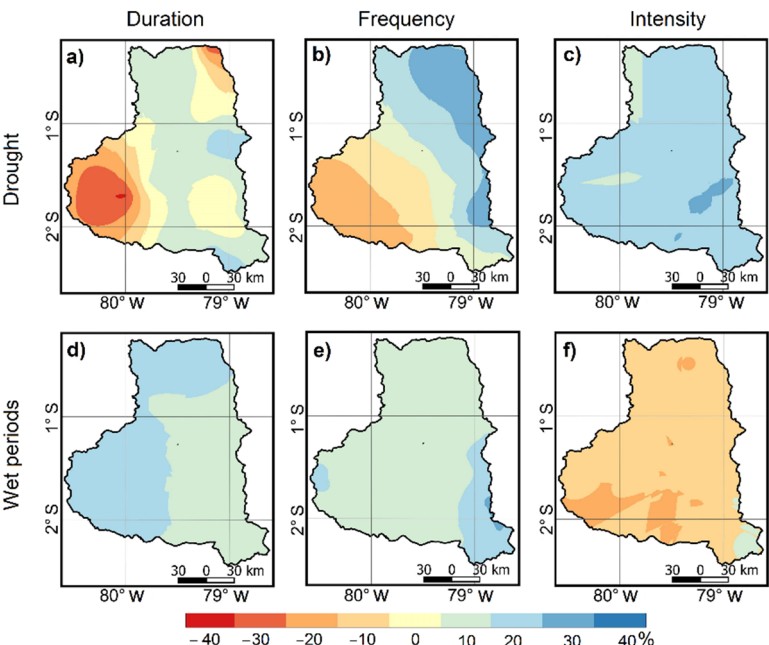

**Figure 9.** Changes (%) in the duration, intensity and frequency of droughts and wet periods for the 2050–2079 period in relation to the 1968–2014 period in the Guayas River basin.

Similarly, an average decrease of ~20% in the frequency of droughts will be expected in the lower basin (coastal plains). In comparison, an increase is shown in the upper basin (~15%) (Figure 9b). The intensity of drought is expected to increase (~20%) across the basin (Figure 9c). Wet periods are expected to increase in duration (~15%) and frequency (~10%) but decrease in intensity (~12%), especially in the western parts of the basin (Ecuadorian coast) (Figure 9d–f).

### 3.4. Hydrological Modeling of the Daule River

The streamflows simulated by GR2M for the calibration and validation phase reveal a good fit, adequately representing the streamflow regime of the Daule River (Figure 10a). The maximum capacity (X1) increased from 7.08 to 7.23 mm, and the exchange parameter (X2) increased from 0.88 to 1.20 mm during validation stage. However, the coefficients of determination ($r = 0.87$; $p < 0.01$) and Nash decreased (Figure 10a); this can be attributed to inadequate representation of GCM datasets for estimating extreme streamflows (Figure 10a). The decrease in the GR2M model's performance has also been observed in other studies [64].

The evaluation of streamflows estimated for the future (2050–2079) by the GR2M model for the Daule river, using evapotranspiration calculated by the method [66] and precipitation from BNU-ESM, CSIRO-Mk3-6-0, FGOALS-g2 and inmcm4 models, shows a strong decrease for the low water period. The wet season, however, will be marked by an increase in the streamflow of the Daule River (Figure 11).

The increase and decrease in streamflows are consistent with the temporal distribution of precipitation and the increase in evapotranspiration, associated with the increase in temperature (see Figure 5a). That is, an average of a 7% increase in precipitation could result in a 69% increase in river streamflow during flooding periods; conversely, an 8% decrease in precipitation could result in a 30% reduction in streamflow for the dry season.

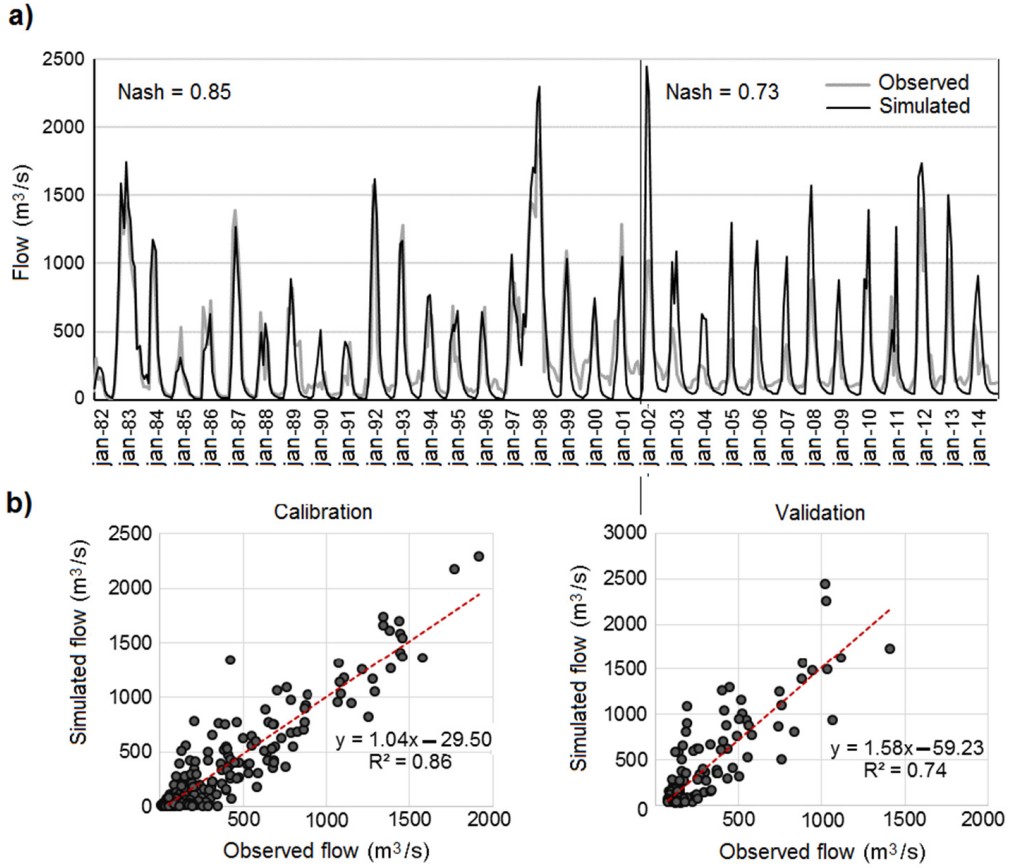

**Figure 10.** GR2M modeling for the Daule River, Guayas River sub-basin, for the calibration (1982–2001) and validation (2002–2014) periods. (**a**) Hydrograph of observed and simulated flows. (**b**) Correlation.

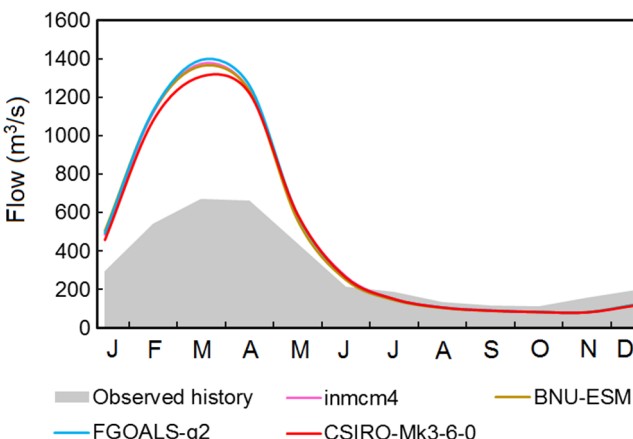

**Figure 11.** Monthly flow distribution of four RCP8.5 climate models predicted for 2050–2079, with respect to the observed flow distribution (1968–2014) in the Daule River, Guayas River basin.

## 4. Discussion

The results of this study predict a warmer climate (~2 °C) for the Guayas River basin. These increases are consistent with a study conducted for a basin of the Ecuadorian Pacific [73]. The temperature increase in this area could accentuate the heat effect in cities with high population density and high development of the construction sector, such as Guayaquil and Durán, causing greater demand for cooling of buildings and, therefore, greater electricity consumption [74,75]. The basin has high potential for agriculture [36], but the increase in temperature could affect the growth and production patterns of livestock because of water stress [76]. It could also affect crop yields because of a shortage of water caused by greater atmospheric evaporation, which, in turn, would result in an increase in soil water stress [61] and a reduction in surface and groundwater [77].

This study also shows an increase (7%) and decrease (8%) in precipitation for the wet (flood) and dry periods, respectively. In the flood period, this increase would cause high soil erosion in the upper and middle parts (mountainous region) and sedimentation problems in the lower part of the basin because of the erosive capacity of rainfall in this area [40]. An increase in rainfall during the wet period could affect the cost of agricultural production by making drainage works necessary for banana and sugarcane crops. Meanwhile, a precipitation decrease in the dry period would affect the availability of irrigation water for water-intensive crops, such as rice; this is also true for rain-fed agriculture.

Our findings suggest that future droughts will be a problem because of an increase in the duration (5%) and frequency (15%) of droughts around the foothills of the western Andes and part of the Ecuadorian coastal plains. This could further damage productive development, because droughts currently constitute one of the main risks for agricultural and livestock production in the basin [78].

Special consideration must be given to the areas determined to be at moderate to high risk so as to manage their water resources and drought in those areas [79]. Drought-induced restrictions on water use can cause a decrease in crop yields (direct impact), triggering an increase in food costs (indirect impact) [80,81].

On the Ecuadorian coast, wet periods would show a greater effect due to the increase in duration (~15%) and frequency (~10%). This area has already been affected by human, economic and infrastructure losses caused by floods in 1965, 1972–1973, 1982–1983, 1987, 1992 and 1997–1998 [82]. The increased frequency, duration and intensity of both wet periods and droughts create an imbalance between atmospheric water supply and demand [83–86]. Increased exposure to extreme events, such as droughts and wet periods, under the influence of a warmer climate, is largely consistent with previous studies in various parts of the world [2,3,87].

Another future impact of climate change in the study area will be changes in the flow regimes of the Daule River. A likely increase in evapotranspiration, due to the temperature

increase, would result in a decrease in streamflows during the dry season (30%), although an increase in precipitation (7%) would trigger an increase (69%) in streamflows for the flood period. The net result could be a significant reduction in river streamflows during the dry season as a result of climate change [88]. This would trigger an increase in demand for irrigation water in the future [89], especially in productive areas like the Guayas River basin. At the same time, the increased flow during the flood season could affect the functioning of dams and reservoirs [90,91], such as the Daule–Peripa, which is located in the middle part of the basin. Changes in flood and low-water streamflows could be a particularly important impact of climate change in terms of danger to people and the environment [91,92].

This study confirms that the use of selected CMIP5 models can provide a robust assessment of climate change impacts on the precipitation regime in equatorial regions. Nonetheless, it highlights the large uncertainty of future temperature and precipitation in GCMs under a high emissions scenario (RCP8.5) [93,94]. There are additional challenges due to the complexity of droughts [95,96] and GR2M model performance associated with input data uncertainty [62]. Ecuador's environment should also be considered a special case because of the country's location, as it is influenced by the Humboldt current, the El Niño phenomenon, trade wind dynamics, the position of the intertropical convergence zone and the presence of the Andes mountain range [36,97–100], which together give this area great climatic variability.

## 5. Conclusions

In this study, a rigorous process was used to select the BNU-ESM, CSIRO-Mk3-6-0, FGOALS-g2 and inmcm4 global climate models for the variables precipitation and temperature, based on the evaluation of 39 phase 5 climate models (CMIP5). These models were selected to investigate the impacts of climate change by representing the climate dynamics of the Guayas River basin in Ecuador.

The monthly mean results for the four models project a ~2 °C increase in temperature and 6% increase in precipitation for the future (2050–2079) relative to the baseline (1968–2014). The monthly distribution of precipitation provides evidence that the temperature increase will be greatest in the months with the lowest precipitation. These dry conditions can increase the risk of forest fire.

The projected relative increase in rainfall amounts is related to the greater duration and frequency of wet periods. The areas likely to be most affected are located in the great plains of the Ecuadorian coast; this will contribute even more to sedimentation in this region. Droughts are likely to be more intense and frequent, particularly along the western range of the Andes and the coastal plains of Ecuador.

Changes in the hydrological regime of the Daule River are also expected; a 69% increase in the streamflow for flood periods is identified from Global Climate Models. This probably is due to increased precipitation. In contrast, a decrease of 30% in the low water level was found because of the potential increase in evapotranspiration caused by higher temperatures. These results indicate the likelihood of problems such as a water deficit during dry periods in the upper part of the basin and increased rainfall and streamflow during flood periods in the lower part of the basin. This information should be considered by stakeholders and policy makers involved in water management when formulating mitigation and adaptation policies to cope with future climate change.

**Author Contributions:** Conceptualization, M.I.-Y., F.I. and R.Z.; methodology, M.I.-Y., F.I., R.Z. and M.G.-M.; analysis, M.I.-Y. and R.Z.; writing—original draft preparation, M.I.-Y.; writing—review and editing M.I.-Y., F.I., R.Z., M.G.-M. and P.C. All authors have read and agreed to the published version of the manuscript.

**Funding:** This research received no external funding.

**Institutional Review Board Statement:** Not applicable.

**Informed Consent Statement:** Not applicable.

**Data Availability Statement:** Publicly available data were analyzed in this study and can be found at the following link: (https://climexp.knmi.nl/plot_atlas_form.py, accessed on 2 October 2020).

**Acknowledgments:** The authors are grateful to the National Institute of Meteorology and Hydrology (INAMHI) in Ecuador for providing temperature and precipitation data (https://www.serviciometeorologico.gob.ec/, accessed on 2 February 2015). The authors also wish to thank the Coupled Model Intercomparison Project (CMIP5) for providing 39 global circulation models (https://climexp.knmi.nl/plot_atlas_form.py, accessed on 2 February 2015). Mercy Ilbay-Yupa thanks the Water Resources Management Group of the Technical University of Cotopaxi (UTC).

**Conflicts of Interest:** The authors declare no conflict of interest.

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
