# Peer review of "Impacts of Climate Change on the Precipitation and Streamflow Regimes in Equatorial Regions: Guayas River Basin"

_water, doi:10.3390/w13213138_

Round 1

Reviewer 1 Report

This paper is part of the multitude of scientific papers that try to analyze the impact of climate change on some elements such as atmospheric precipitation. The choice of a representative area of the equatorial area, Guayas River Basin, and the use of g 39 CMIP5 models under the RCP8.5 emissions scenario facilitated the achievement of results that can be considered a model of analysis of climate changes in basin-scale studies in equatorial regions.

Some minor modification the authors must made before publishing this paper:

  1. the text from row 78 to 82 ”The Guayas River basin is  one of the most important in Ecuador and is home to approximately 40% of the popula80 tion. This area has been impacted by severe and prolonged extreme events that have had destructive effects on the economy, food security, infrastructure and ecosystems [41- 43].” can be moved in upper part of the introduction.
  2. in table 1 the authors must put 19 before 68-2014....1968-2014
  3. table 2 is not necessary
  4. in row 180 to 181 appear some errors: (equation ¡Error! No se encuentra el origen de la referencia.)
  5. attention to the figure number row 369 Figure 1314!, row 397 figure 1920!.
  6. in figure 1920! the observed values can be represented with smooth line

Reviewer 2 Report

The manuscript presents a standard assessment of climate change impacts on the hydrology of Guyas River basin. As it is similar to many recent papers, its novelty is limited. However, the research is generally well designed and the manuscript is well organized. The use of Taylor's diagram is interesting and not common in similar published papers.

My major criticism regards the selection of separate models to describe the impacts on precipitation and the impacts on temperature (lines 293-297). Temperature and precipitation are closely related and the scenarios that consider the temperature projection of one model and the precipitation of another model are not consistent. The authors should select a set of models based on their overall ability to reproduce the historical climate. Moreover, the discussion of the results analyzes how precipitation changes vis-a-vis the changes in temperature. This makes no sense if the projections come from different models. The projection of streamflow is also inconsistent.

Authors should not try to identify the "single best model". By selecting several models an assessment of the uncertainty associated with these projects is obtained.

As there are many similar papers, some parts of the manuscripts can be eliminated to shorten the manuscript. I suggest removing or abbreviating the following parts:

  • paragraph 46-53, on the definition and characterization of droughts
  • figures 3 and 4
  • lines 207-222 on SPI, an indicator which is extensively described in the literature

The manuscript title should be changed, as the work also evaluated the impacts of climate change on streamflow, and not only on precipitation.

In line 76, the authors claim that there are no basin-scale studies in equatorial regions. I find it hard to believe.

In lines 229-234 the authors must clarify the definition of drought duration and intensity, based on the concept of drought. A drought is a period of successive months where SPI is lower than a certain value. Is there a minimum number of successive months to declare a period as a drought? Some authors only consider a drought when there are 3 consecutive months when SPI is negative. Then duration is the average number of months of all droughts. The intensity is the average value of SPI during all the droughts.

Line 280: the performance assessment of the GCMs is not done for RCP8.5. It is performed using the control run.

Line 299 - How does figure 8 show that the selected models represent well the annual cycle of temperature and precipitation if the historical means are not presented?

A thorough language review is recommended, as there are parts in the text which are badly stated or confusing. A few examples:

  • line 32: Climate change plays an important role in society and the economy?
  • line 34: "would lead"??
  • line 42-44:
  • line 73: many countries are experiencing climate change:  
  • lines 116-117

Also, some examples of typos that should be corrected:

  • The figure numbering is not correct
  • Table 1: Longitude, not length.
  • line 157: and not y
  • line 280: GCMs not CMGs

The discussion of the results should be completely reviewed given my above comments. 

I, therefore, recommend rejecting the paper and inviting the authors to submit a reviewed version. 

Reviewer 3 Report

The paper is addressed to explore future projections of both monthly  temperature and rainfall amount  in Guayas River basin ( Ecuador) under global warming scenarios. These climate variables  are  inputs of a hydrological model able to reconstruct the streamflow of the river Daule.  The paper is an other of the many papers that using model simulations by CMIP5 try to extract relevant hydrological information for specific geographic areas. The methodology is almost typical; a) download GCM simulation for the considered region; b) compare simulations with the observations; c) apply some kind of bias corrections; d) reconstruct projections bias corrected; e) use the projection as input for  hydrogical models. 
Personally I am rather critic on the use of bias correction methods to fit simulations to observations. I never understood how in the contest of complex, caotic and nonlinear climatic system bias correction methods make sense. However the scientific community seems agree with the use of such methods and thus this criticism is not part of the paper judgement.  Globally the paper is well written, authors should just control some typing errors in the text and relationships.  The study also is well conducted, methodology is clearly explained, results are consistent. Only my concern is how the averaged temperature and precipitation over the region are obtained by simulations. Given the reduced extention of the area I believe the just few points of GCM simulation grid are within that area. This question would deserve to be discussed. 
An other criticism is the use of only the RCP8.5 among the other mitigation scenarios of CMIP5. This choice would make sense if authors had proposed a new methodology and use the RCP8.5 as a test to validate the method. Since there is not a particular methodology originality in the paper  probably the use of more mitigation scenarios and an analysis of the incertainties in projections,which is the crucial point of any future projection,  would improve the quality of manuscript.  However  the manuscript, also in the present form, could be taken in consideration for pubblication since it provide some useful information for a specific geographic ragion that could be useful for Water readers. 

Author Response

Consulte el archivo adjunto

Round 2

Reviewer 2 Report

My main objections to the original report have been addressed. However, an extensive English review is recommended as the text is not rigorous is many parts. A few examples (among many) are referred below:

The added text in the introduction is confusing. 

Line 155: "The selection of the best temperature and precipitation GCMs in comparison to observed dataset was performed using the skill score (S) for each model." - what is the "the best temperature and precipitation GCM"?

Line 215: "SPI-1 for drought and wet periods was characterized by three main aspects: intensity, duration and frequency [3,60,61]. " - I believe the authors mean that the drought regime was characterized by the frequency, duration, and intensity of droughts computed from the SPI-1.

Line 219: "In addition, the frequencies of drought events according to their duration (1 – 5 months) were also calculated for periods greater than or equal to 30 years". - What do the authors mean and how were these estimates computed?

It is not clear how table 2 values are used. These threshold values may be used to assign a category to a given month. As drought is defined by a period of several consecutive months below a threshold, how is each drought categorized?

Line 267: "The 39 GCMs of CMIP5 phase 5 under the RCP8.5 scenario were evaluated." - My comment was not addressed. The performance of a GCM is evaluated by comparing its results for the control run with observed values. Why is the RCP8.5 scenario referred here?

Figure 6 and 7 present cdf of drought duration and intensity. A similar graph on drought frequency should be presented.

Figure 6 is missing a legend. The accompanying text discusses the duration of severe and extreme droughts. It is not clear how the figure justifies this distinction between severe and extreme droughts.

The same comment applies to Figure 7. The discussion refers to severe and extreme droughts and is not clear how the figure justifies this distinction.

Figure 8 is confusing.

Are the maps of Figure 9 presented in %?

Author Response

Thank you for your helpful comments and suggestions, which have been very useful for improving the manuscript. Your comments have been taking account point by point.

Round 3

Reviewer 2 Report

The manuscripts has been improved and most of my criticisms have been addressed. However the text is at times confusing and not accurate. A few examples are presented below:

Abstract – What that the following sentence mean: “The climate projection based on the four rigorously selected models represents the climate dynamics of the study area”

Line 51 – Rewrite: “Droughts characterized can severely alter the hydrologic regime [21], affect river water quality and impact aquatic ecosystems”.

Line 133: The verb is missing in iv)

Line 211-214 – Why is there a reference to 1-5 dry months in the definition of drought duration? This definition may lead to many consecutive droughts separated by a short period, when in fact it is a single drought. A warning should be added to the text.

Figure 8 – I would not represent future wet periods and droughts by a continuous black line; it suggests there is continuity from extreme drought to extremely wet, which is not the case. A set of horizontal bars is probably better.

I recommend a thorough revision of the text by an English native speaker.

Author Response

Gracias por sus valiosos comentarios y sugerencias, que han sido muy útiles para mejorar el manuscrito. Tus comentarios se han ido tomando en cuenta punto por punto. (Thank you for your valuable comments and suggestions, which have been very helpful in improving the manuscript. Your comments have been taken into account point by point.)
